

# Submarine melt rates and mass balance for Greenland's remaining ice tongues

Nat Wilson[1,2], Fiammetta Straneo[3], and Patrick Heimbach[4]

[1]MIT-WHOI Joint Program in Oceanography/Applied Ocean Engineering, Cambridge, Massachusetts, USA
[2]Geology and Geophysics Department, Woods Hole Oceanographic Institution, Woods Hole, Massachusetts, USA
[3]Physical Oceanography Department, Woods Hole Oceanographic Institution, Woods Hole, Massachusetts, USA
[4]Jackson School of Geosciences and Institute for Computational Engineering and Sciences, University of Texas at Austin, Austin, Texas, USA

*Correspondence to:* Nat Wilson (njwilson23@gmail.com)

**Abstract.** Ice shelf-like floating extensions at the termini of Greenland glaciers are undergoing rapid changes with potential implications for the stability of upstream glaciers and the ice sheet as a whole. While submarine melting is recognized as a major contributor to mass loss, the spatial distribution of submarine melting and its contribution to the total mass balance of these floating extensions is incompletely known and understood. Here, we use high-resolution WorldView satellite imagery collected between 2011–2015 to infer the magnitude and spatial variability of melt rates under Greenland's largest remaining ice tongues, Nioghalvfjerdsbræ (79 North Glacier), Ryder Glacier, and Petermann Glacier. Submarine melt rates under the ice tongues vary considerably, exceeding $50\,\mathrm{m\,a^{-1}}$ near the grounding zone and decaying rapidly downstream. Channels, likely originating from upstream subglacial channels, give rise to large melt variations across the ice tongues. We compare the total melt rates to the influx of ice to the ice tongue to assess their contribution to the current mass balance. At Petermann Glacier and Ryder Glacier, we find that the combined submarine and aerial melt approximately balances the ice flux from the grounded ice sheet. At Nioghalvfjerdsbræ the total melt flux ($14.2\pm1.6\,\mathrm{km^3\,a^{-1}}$ water-equivalent) exceeds the inflow of ice ($10.2\pm0.59\,\mathrm{km^3\,a^{-1}}$ water-equivalent) indicating present thinning of the ice tongue.

## 1 Introduction

Mass loss from ice sheets is often greatest at the marine termini (Truffer and Motyka, 2016). Here, ice shelves and ice tongues are hypothesized to influence the stability of the upstream glaciers, and thus of the entire ice sheet (e.g. Dupont and Alley, 2005; Furst et al., 2016). For these reasons, monitoring and understanding processes at marine outlets is key to understanding past and predicting future ice sheet variability.

In Greenland, floating ice tongues currently protrude, or have recently protruded, from the termini of several major outlet glacier systems. However, warming air and ocean temperatures since the mid-1990s have accompanied the reduction or disappearance of most of Greenland's floating ice tongues. This includes the rapid retreat and collapse of the Jakobshavn Isbræ beginning in 1998 (Motyka et al., 2010). Beginning in 2012, the floating ice tongue of Zachariæ Isbræ, one of the largest in Greenland, has been in a phase of retreat and collapse (Mouginot et al., 2015). Partial loss of the ice tongue has occurred at



Petermann Glacier in northwestern Greenland via a series of major calving events (Falkner et al., 2011; Münchow et al., 2014), and Mouginot et al. (2015) and Kjeldsen et al. (2015) have reported thinning at Nioghalvfjerdsbræ in northeast Greenland. In all cases, changes in submarine melting have been identified as the likely driver of these ice tongue changes (e.g. Holland et al., 2008; Motyka et al., 2011; Münchow et al., 2016).

Unfortunately, accurately assessing the submarine melt terms in the mass balance of ice tongues is challenging. This is partly due to the difficulty in making extensive in-situ measurements (Straneo et al., 2012a), and partly due to the trade-offs in spatial and temporal resolution made by existing remote measurement platforms. However, given the recent changes in ice tongues and ice shelves in Greenland and Antarctica and the role they have in buttressing upstream ice flow (Dupont and Alley, 2005), understanding the spatial variability and magnitude of submarine melting is needed to predict future ice sheet variability and

sea level rise (Joughin et al., 2012).

We address the observational gap by estimating current melt rates under Greenland's three largest remaining ice tongues: Nioghalvfjerdsbræ (79N), Ryder Glacier (RG), and Petermann Glacier (PG). 79N is the largest existing ice tongue in Greenland by area with a 65 km-long floating ice section confined within a 20 km-wide fjord (Fig. 1) and is grounded at a depth of 700m (Mayer et al., 2000). It is one of three outlet glaciers of the Northeast Greenland Ice Stream (Fahnestock et al., 1993), together

with neighbouring Storstrømmen Glacier and Zachariæ Isbræ. While Zachariæ Isbræ has undergone both recent collapse and acceleration, change at 79N has been less visible with no major calving events despite high melt rates first inferred by Mayer et al. (2000). Recently though, several studies have reported that thinning at 79N appears to have taken place during the 21st century (Kjeldsen et al., 2015; Mouginot et al., 2015).

In northwestern Greenland, RG is an ice tongue roughly 25 km long within a 10 km wide fjord and grounded around 500 m

depth (Joughin et al., 1999). RG episodically experiences both large calving events and "surge" episodes (Joughin et al., 1996). Further west, PG is the second largest ice tongue in Greenland by area. PG also experiences episodic major calving events,(Falkner et al., 2011) with recent examples from 2001, 2008, 2010, and 2012. The grounding line depth at PG is approximately 500 m (Johnson et al., 2011), and rapid basal melting was noted by Rignot (1996). For both 79N and PG, steady state estimates suggest that the ice tongue's mass balance is dominated by submarine melting (Mayer et al., 2000; Johnson

et al., 2011; Falkner et al., 2011; Münchow et al., 2014).

Recent estimates of melt rates for these ice tongues are derived from flux gate or flux divergence methods (e.g. Rignot and Steffen, 2008; Enderlin and Howat, 2013). These approaches rely on an assumption of steady state, in which the ice tongue is neither thinning or thickening. Münchow et al. (2014) use ICESat and IceBridge data to show that non-steady melt was important at PG prior to calving. Furthermore, while broad generalizations of the melt rate patterns are currently available

from Mayer et al. (2000) and Seroussi et al. (2011) there are no published efforts to carefully map the distribution of melting at 79N and RG over the entire ice tongue.

To address this, we compute Lagrangian hydrostatic ice thickness change over the period 2011–2015 from digital elevation models (DEMs) constructed from WorldView satellite imagery. The results from this analysis allow us to both map the spatial variability of melting and to determine the current ice tongue mass balance. This method has been applied in Antarctica (e.g.

Dutrieux et al., 2013; Moholdt et al., 2014), but never before to Greenland's ice tongues. Contemporary velocity estimates



come from optical feature tracking applied to the same imagery used for the elevation models. We combine the material ice thickness change with flow divergence fields to compute the sum of surface and submarine ice tongue melt rates, accounting for dynamic thinning. The surface component of the transient melt, estimated from 0.76–2.0 m of water-equivalent depending on the ice tongue, was extracted from version 2.3 of the Regional Atmospheric Climate Model (RACMO2.3, Noël et al., 2015)

and subtracted from the net melt rate to yield estimated submarine melt rates. We present the melt rates as a temporal average over the years for which data are available. Compared to previous approaches that use flux-gate approaches and rely on steady state assumptions, this method more easily incorporates transient thinning or thickening of ice tongues.

## 2  Methodology

### 2.1  Data sources and preparation

We used optical image pairs from WorldView-1, WorldView-2, and WorldView-3 satellites, collected over the period 2011–2015 and distributed by the Polar Geospatial Center at the University of Minnesota. We orthorectified these images with the Greenland Ice Mapping Project Digital Elevation Model (Howat et al., 2014) and then constructed new DEMs using the open-source NASA Ames Stereo Pipeline software (Shean et al., 2016). Vertical and horizontal errors in the DEMs are reduced by re-referencing non-glaciated regions to a composite DEM constructed by averaging all available DEMs. Shean et al. (2016)

demonstrates that errors in WorldView DEMs are highly correlated, such that co-registration to a common data source removes much of the global uncertainty.

Although the accuracy of existing tide models is in most cases unknown, we subtract the tidal elevation predicted from the AOTIM5 tide model (Padman and Erofeeva, 2004). The distance between the most landward edge of the grounding zone and the point where the ice tongue is in hydrostatic equilibrium is on the order of a few kilometers (Brunt et al., 2010). Therefore,

the influence of tides will be fully reflected in the ice tongue elevation beyond a narrow buffer zone at the grounding zone, which we exclude from the full analysis (see below). Comparison between the AOTIM5 estimates and a short record collected in 2009 at 79N (G. Hamilton and L. Stearns, personal communication, 2015) reveals an offset in the tide model of approximately 0.2 m.

### 2.2  Velocity

Over an ice tongue, where basal stresses are negligible and ice aspect ratio is small, surface velocity is close to the depth-

averaged velocity. We compute a grid of correlation offsets by comparing image chips extracted from overlapping hillshaded maps computed from each DEM. We compute the cross-correlation between extracted image chips and then use a Gaussian sub-pixel peak-finding calculation (Debella-Gilo and Kääb, 2011) to estimate displacement. The cross correlation at the peak normalized by the cross correlation standard deviation is retained as a measure of correlation quality. To reduce the area that must be searched for correlation over long temporal baselines, we estimate feature displacement from the MEaSUREs ice

velocity dataset (Joughin et al., 2010) and extract image chips representing a small neighbourhood around these estimates.



## 2.3 Melt rates

Mass conservation of an ice tongue requires that

$$\frac{\partial h}{\partial t} + \nabla \cdot (\boldsymbol{u}h) = \dot{a}, \tag{1}$$

where $h$ is the ice tongue thickness, $\boldsymbol{u}$ is the depth-averaged velocity, and $\dot{a}$ is the rate of ice thickness change due to combined
surface and submarine mass balance. Ice thickness is inferred by assuming hydrostatic equilibrium over the floating ice tongue.
Hydrostasy is a good approximation over sufficiently long horizontal length scales and shallow tongue thickness gradients
(Brunt et al., 2010). Immediately downstream of the grounding line the hydrostatic assumption is less justified, so we exclude
data within a few kilometers of the grounded ice sheet. The BedMachine mass-conserving bed elevation dataset (Morlighem
et al., 2014) serves as a guide to identify where the ice reaches flotation.

By neglecting the time derivative in ice thickness, the steady state melt rate can be estimated from the flux divergence alone.
For example, Seroussi et al. (2011) and Enderlin and Howat (2013) have applied this method to Greenland's marine terminating
glaciers. To include the time derivative in (1) and, in so doing, account for dynamic thickening or thinning of the ice tongue, we
compare pairs of ice thickness estimates separated by a temporal baseline. If an Eulerian reference frame is used, this approach
is susceptible to temporal aliasing of elevation variations advected along with ice flow. In a Lagrangian framework, temporal
aliasing can be avoided as done previously with satellite altimetry data (Moholdt et al., 2014) and stereogrammetric elevation
maps (Dutrieux et al., 2013). In this case, the mass balance equation is written in terms of the material time derivative,

$$\frac{Dh}{Dt} + h\nabla \cdot \boldsymbol{u} = \dot{a}. \tag{2}$$

The Lagrangian framework has the additional advantage of avoiding explicit calculation of an ice thickness derivative. Scenes
for correlation and melt rate extraction are selected by searching the list of available DEMs for pairs that overlap by minimum
of $50 \, \text{km}^2$ and that are separated in time by between 180 and 600 days. We compute a temporal mean by first binning average
estimates of $Dh/Dt$ by month in order to offset bias due to the optical imagery being more available during seasons with more
daylight.

## 2.4 Grounding line fluxes

We estimate the flux of ice from the grounded ice sheet into the floating ice tongue from the correlation-derived ice velocity
and ice tongue thickness estimates over a flux-gate at the downstream end of the grounding zone. Retaining the assumption
from above that glacier velocity is invariant with depth, the total grounding line flux $\Phi$ is

$$\Phi = \int_a^b \hat{u}(x)h(x)dx, \tag{3}$$

where $\hat{u}$ is the grounding line-normal velocity and limits $a$ and $b$ are as close as possible to the glacier margins. We note that our
results are more conservative than other estimates made further upstream, because some ice has melted by the time it crosses
the upstream gate.



### 2.5 Errors and uncertainties

Uncertainty in the time-averaged melt estimates that we derive is due to measurement error, DEM co-registration and correlation errors, errors associated with the tide-corrected ocean elevation, and temporal variability in the ice tongue surface elevation due to accumulation. We make the assumption that these effects are unbiased and attempt to quantify the combined

effect of these error sources using a jackknife resampling scheme. This provides uncertainty estimates for the computed melt rates and submarine melt fluxes. As the melt rates are generated from a large number of WorldView DEM pairs, we generate new melt rates by excluding individual DEMs and recomputing the relevant measures using the remaining subset of DEMs. This accounts for sources of random measurement error. Uncertainties reported in the text and figures represent one standard deviation around the computed mean value. Uncertainty in the flux gate volumes is expressed by including the variance from

the full set of generated DEMs in the calculation of ice thickness at the flux gate. Some additional error is due to non-random effects such as downstream firn densification, which would yield melt overestimates. We expect this to be small as the ice tongues considered are well below the equilibrium line and any remaining firn layer is expected to be thin.

The spatial scales at which ice thickness variability (and melt rate variability) may be diagnosed from surface elevation data is physically limited by the melting time scale and ice viscosity. To characterize the accuracy of our ice thickness estimates

empirically, we have compared the hydrostatic ice thickness to radar measurements from the MCoRDS echo sounder flown as part of the Operation IceBridge program from 2010–2012 (Leuschen et al., 2010). These data are limited to only a few lines over the ice tongues, but match the ice thickness inferred from the DEMs with a coefficient of determination of 0.93. We expect errors due to non-hydrostatic portions of the ice tongue to be small away from the grounded ice when averaged over coarse grids as in Figure 1 or the entire ice tongue, as in Table 1.

## 3   Results

Melt rates for the three ice tongues indicate that the largest melt rates occur near the grounding line (Fig. 1) and decay within 10 km down-glacier from the upstream boundary of our analysis. Maximum melt rates at 79N drop from 50–60 m yr$^{-1}$ near the grounding line to 15 m yr$^{-1}$ 15 km downstream (Fig. 2). Further downstream, melt rates at 79N drop to near zero. Slightly lower grounding zone melt rates are obtained for Petermann (40–50 m yr$^{-1}$) decreasing to a background rate of 10 m yr$^{-1}$ over

a distance of 15–20 km (Fig. 3). At RG, maximum melt rates near the grounding line are similar to those observed at PG; melt rates away from the grounding zone are in the range of 10–20 m yr$^{-1}$ (Fig. 4). These results are qualitatively consistent with aggregated rates obtained by previous studies (e.g. Mayer et al., 2000; Rignot and Steffen, 2008; Münchow et al., 2014).

In addition to this general decrease of melting away from the grounding zone, our results also show smaller scale variations in melt rate on the scale of a few kilometers. At 79N, the fastest melting occurs near the southern two-thirds of the grounding

zone and lower along the northern third (Fig. 2). The largest melt rates correspond to the thickest and fastest moving part of the ice stream. Large variability in melt rates in the across-tongue direction are also observed near the grounding zones of RG and PG. In particular, large melt rates (in excess of 50 m yr$^{-1}$) near the grounding zone of RG occur at the head of an incised channel associated with an elongate 15–20 m depression in surface elevation (Figs. 1, 4). Melt rates on the west side of the





grounding zone at RG are much lower in spite of similar ice thickness and draft. Recent bed elevation inversions by Morlighem et al. (2014) indicate that the region of rapid melting identified at RG is at the end of a bedrock trough that reaches over 100 km upstream beneath the Greenland ice sheet. Similarly, the melt rates patterns at PG appear to vary on a scale similar to the channelised geometry of the ice tongue base (Fig. 3), consistent with the broad spatial patterns derived from a steady-state ice

flux divergence approach (Rignot and Steffen, 2008).

The annual surface melt rate predicted at 79N by RACMO2.3 over the period 2011–2015 is $1.5\,\mathrm{m\,a^{-1}}$ water-equivalent (w.e.). Multiplying this by the ice tongue surface area of $1600\,\mathrm{km^2}$ suggests that submarine melting accounts for approximately 80% of the annual non-calving mass loss at 79N. We compute the ice volume flux passing from the grounded ice sheet through the point where the ice tongue becomes hydrostatic to be $10.2\pm0.59\,\mathrm{km^3\,yr^{-1}}$ w.e. This may be smaller than the $12\pm1\,\mathrm{km^3\,yr^{-1}}$

reported by (Rignot and Steffen, 2008) because of our exclusion of a non-hydrostatic buffer around the actual grounding line. We find that the incoming ice volume is 70% of the observed melt flux within the region considered ($14.2\pm1.6\,\mathrm{km^3\,yr^{-1}}$ w.e.), providing direct evidence that the ice tongue is melting at a faster rate than ice is being replenished from upstream (Fig. 5). Excluding calving, the annual net volume loss is 1.3% of the ice tongue's total volume. The imbalance between melting and advective replenishment of ice volume at 79N is consistent with the thinning diagnosed from aerial imagery (Kjeldsen et al.,

2015) and by comparing ice thickness derived from radar measurements during the 1990s and the 2010–2014 period (Mouginot et al., 2015).

By contrast, the grounding line flux at RG of $1.9\pm0.12\,\mathrm{km^3\,yr^{-1}}$ w.e., slightly smaller than a previously-reported (Rignot et al., 1997) value of $2.3\,\mathrm{km^3\,yr^{-1}}$, is not significantly different from the total melt flux ($2.1\pm0.21\,\mathrm{km^3\,yr^{-1}}$ w.e.). Similarly, at PG, which lost substantial area from 2001–2012, the present combined observed melt fluxes ($11.7\pm1.4\,\mathrm{km^3\,yr^{-1}}$ w.e.) are

only slightly larger than the grounding line fluxes ($10.8\pm0.52\,\mathrm{km^3\,yr^{-1}}$ w.e.). Excluding calving, the annual net volume loss is 0.4% of the current ice tongue volume. Based on a conservative estimate of melting under the former terminus of PG of $5$–$10\,\mathrm{m\,yr^{-1}}$ and a calved area of approximately $250\,\mathrm{km^2}$, we speculated that the pre-2010 melt flux may have been around $13\,\mathrm{km^3\,yr^{-1}}$ w.e. As a result, while we do not observe submarine melting at PG to be a driver of substantial net volume loss in its current configuration, it may have been a significant contributor prior to 2010.

In all cases, we find a significant correlation between high melt rates and either ice tongue draft and ice tongue basal slope (Fig. 6). The three ice tongues studied above do differ in how the relationship between melt rates and draft/slope is expressed. At 79N, the relationship with slope is perhaps the weakest of the three, with high melt rates inferred even in regions of low basal slope. In general, most of 79N is very low slope, which appears to correspond with the lowest average melt rates of the three glaciers. Given this, the large mass imbalance at 79N is a result of its large area rather than anomalously high average

melt rates. In contrast, RG and PG have more similar distributions of basal slope.

## 4    Discussion

At all ice tongues the highest melt rates are found near the grounding line. This is consistent with the presence of relatively warm, dense waters of Atlantic origins on the nearby continental shelves and fjords (Straneo et al., 2012b). The in-situ temper-



ature of these waters can exceed 1°C at 79N (Wilson and Straneo, 2015), while those at PG in the northwest are consistently measured below 0.5°C (Rignot and Steffen, 2008; Münchow et al., 2016). To our knowledge, no ocean temperatures have been recorded near RG. Nevertheless, profiles collected in 2006 from the continental shelf 150 km away measured maximum temperatures below 300 m near 0.25°C (Steele, 2016). We expect the melt rate at the grounding depth to vary both with the

water temperature above the freezing point (the thermal forcing), which in turn varies with pressure, as well with factors that regulate the heat transfer across the ice/ocean boundary layer including ice slope, subglacial discharge (Jenkins, 2011; Straneo and Cenedese, 2015). Assuming that the ocean temperature at the grounding depth is equal to that observed in the fjords (and on the shelf for RG) and equal to 1°C, 0.5°C and 0.25°C for 79N, PG and RG, respectively, and grounding depths of 700 m, 500 m, and 450 m, we estimate the thermal forcing for these three systems to be about 3.4°C, 2.7°C, and 2.5°C. In a broad

sense, the melt rates near the grounding zone are consistent with differences in thermal forcing computed at the three ice tongues.

Thermal forcing aside, we also expect higher melting at steeper basal slopes due to the larger entrainment in rising melt water plumes (Little et al., 2009). Again our derived melt rates are largely consistent with this overall pattern but suggest that there is no simple relationship between slope and melt rate.

Our results also indicate that these tongues exhibit a large lateral variability in melt rates due to the presence of basal channels. Channelization is likely derived from heterogeneity in the ice tongue thickness, flow, and oceanic characteristics (Sergienko, 2013). At RG, the deep central channel along the base of the ice tongue appears to be a product of high melt rates, and may also concentrate them in a positive feedback. The lower advection rates at RG may also make it more susceptible to developing deep channels, as the rate of replenishment of ice from upstream is lower. At PG, ice advection rates at the

grounding line are similar to 79N, and so the presence of deep channels must be due to either more intense grounding line melting or more active maintenance by submarine melting along the length of the ice tongue (e.g. Gladish et al., 2012).

## 5   Conclusions

Our results highlight high spatial heterogeneity in melt rates beneath Greenland's ice tongues in both the along-flow and the across-flow directions. This latter component of spatial variability is typically ignored in models, but is likely to be important

to accurately predict ice tongue and buttressing sensitivities to ocean temperature and ice geometry changes. We also show that PG and RG are in their current geometries close to maintaining their total volume, with grounding line influxes balancing inferred melting. We find that in spite of apparently minor changes in surface area in recent decades at 79N, net mass losses there are the highest of the remaining large ice tongues in Greenland. We speculate that major changes will take place at 79N in the future as the ice tongue thins and eventually becomes ungrounded at its terminus. This potentially has important

implications for buttressing of the inland portion of the outlet glacier that is the exit of the Northeast Greenland Ice Stream.



*Code and data availability.*   The satellite data used in this research are available subject to the data licensing requirements of the Polar Geospatial Research Center. Lists of the images used will be provided upon request. Code used to analyze the satellite images is available publicly at https://bitbucket.org/njwilson23/worldview_processing or upon request.

*Author contributions.*   NW developed the code, performed the WorldView data analysis, and was the primary contributor to the text of the manuscript. FS actively contributed ideas and also submitted text in the manuscript. PH provided guidance by contributing helpful advice on the scientific ideas and feedback on the manuscript.

*Competing interests.*   The authors are not aware of any competing financial interests.

*Acknowledgements.*   NW, FS, and PH were supported by NASA NNX13AK88G and NSF OCE 1434041. WorldView-1/2/3 imagery were made available by the Polar Geospatial Center at the University of Minnesota and DigitalGlobe. Gordon Hamilton and Leigh Stearns provided GPS data from 79N. RACMO2.3 data were kindly made available by M. R. van den Broeke. David Shean contributed helpful advice and discussion regarding processing WorldView imagery with the NASA Ames Stereo Pipeline.



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



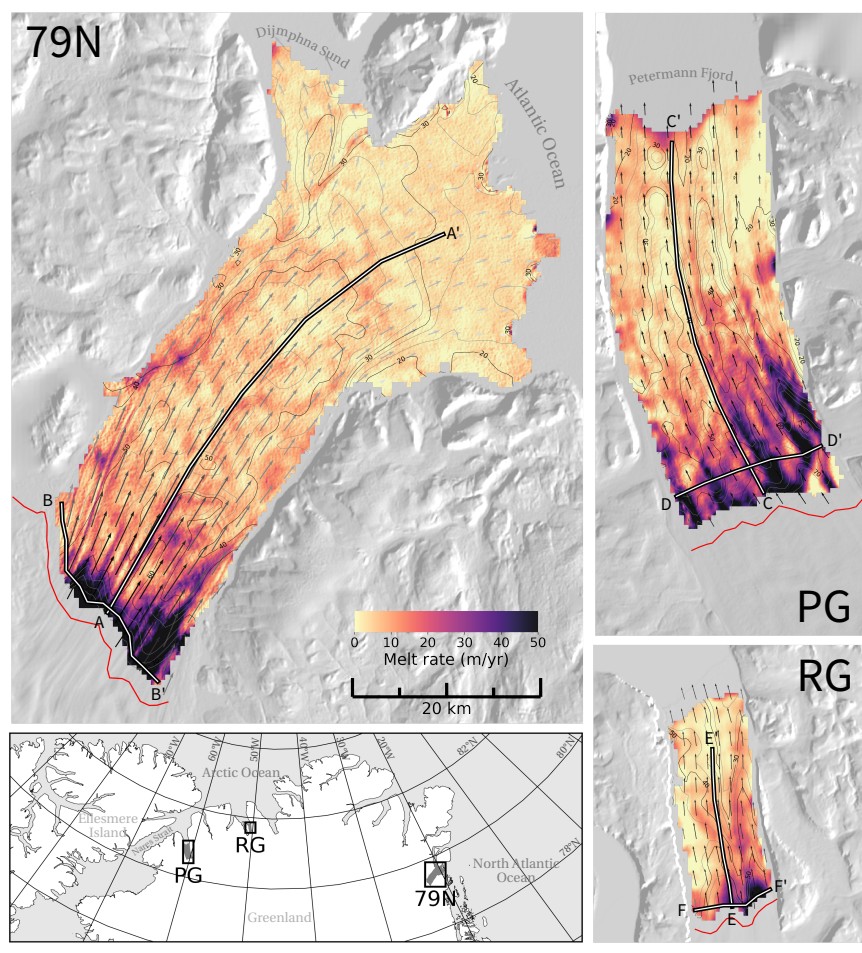

**Figure 1. Map of submarine melt rates from Greenland's major ice tongues derived from WorldView satellite DEMs.** The colour shading shows regions of rapid submarine melting, while the arrows indicate ice flow directions. Red line approximates the grounding line. Inset map in the lower left provides regional context.



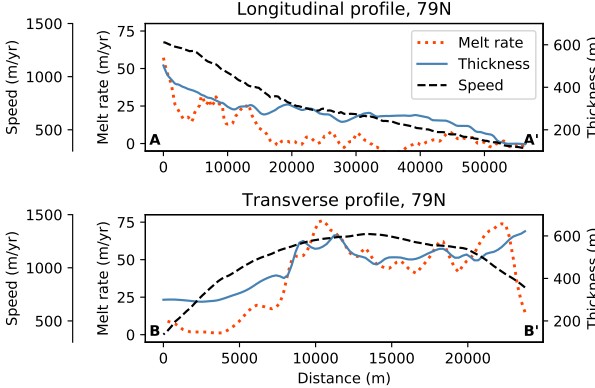

**Figure 2. Velocity, melt rate, and ice thickness profiles for 79N.** Longitudinal profile along the glacier centreline. Transverse profile is taken approximately 1 km downstream of the upstream flux gate.

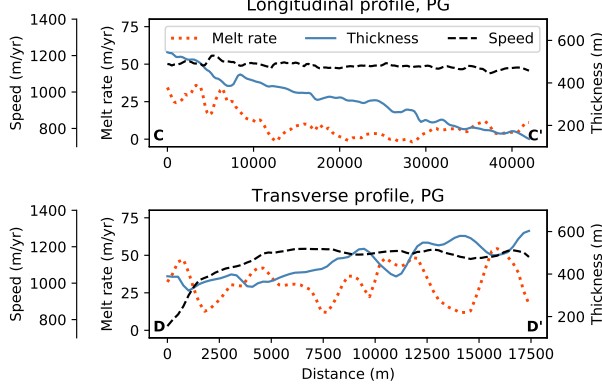

**Figure 3. Velocity, melt rate, and ice thickness profiles for PG.** Longitudinal profile is outside of a long sub-shelf channel. Transverse profile is taken approximately 1 km downstream of the upstream flux gate.



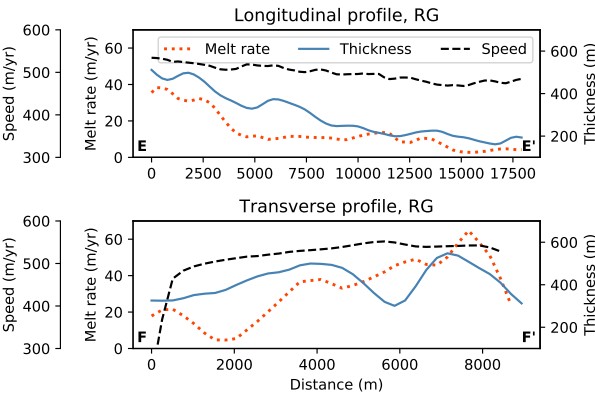

**Figure 4. Velocity, melt rate, and ice thickness profiles for RG.** Longitudinal profile computed along a transect to the west of the major sub-shelf channel. Transverse profile is taken approximately 1 km downstream of the upstream flux gate.



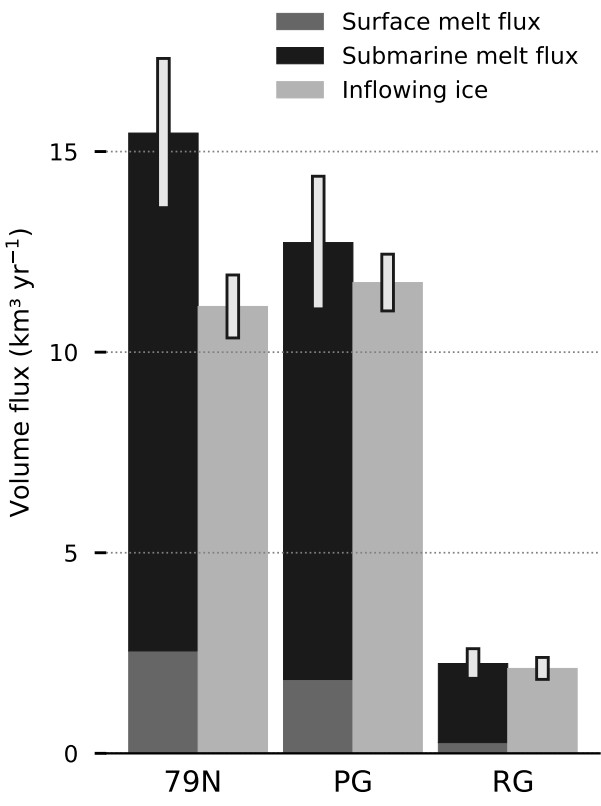

**Figure 5. Comparison of ice volume fluxes.** Melt flux and grounding line volume flux for each floating ice tongue are shown, with melt fluxes partitioned between submarine and subaerial components. Error bars represent one standard deviation above and below the estimate.





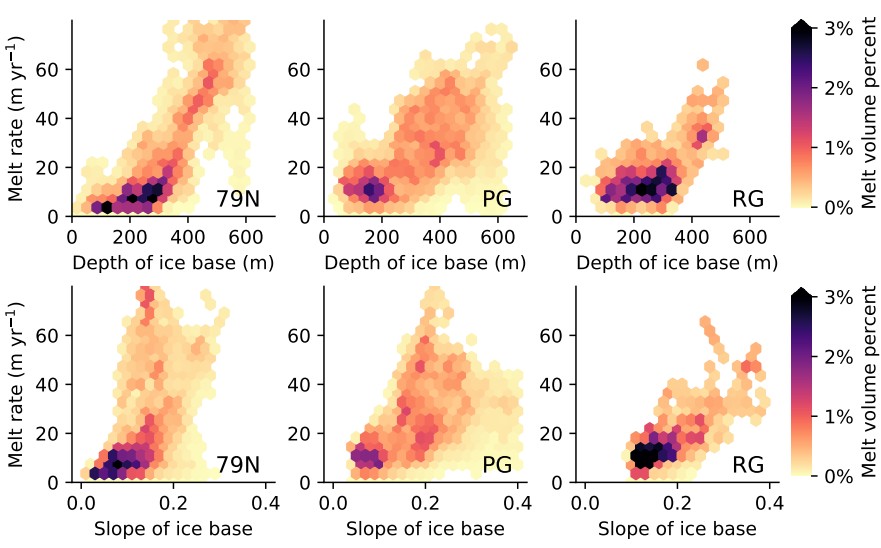

**Figure 6. Relationships between estimated melt rates and the depth and slope of the ice tongue base.** Shading serves as a visual guide to the density of observations in each cell.



**Table 1. Volume fluxes for Greenland's major ice tongues, in freshwater equivalent.** Ranges on submarine melt flux are one standard deviation and derive from surface elevation uncertainty.

| Glacier | Volume km$^3$ | Inflowing ice km$^3$ yr$^{-1}$ | Submarine melt flux km$^3$ yr$^{-1}$ | Surface melt flux km$^3$ yr$^{-1}$ |
|---------|--------|--------------|--------------------|-------------------|
| 79N | 314 | 10.2±0.59 | 11.9±0.96 | 2.3±1.3 |
| RG | 51.0 | 1.9±0.12 | 1.8±0.12 | 0.25±0.17 |
| PG | 215 | 10.8±0.52 | 10.0±1.2 | 1.7±0.68 |