# Peer review of "Satellite-derived submarine melt rates and mass balance (2011–2015) for Greenland’s largest remaining ice tongues"

_The Cryosphere, 2017_

## Referee Comment (RC1) · Anonymous Referee #1 · 29 Jul 2017

The manuscript entitled "Submarine melt rates and mass balance for Greenland's remaining ice tongues" by Wilson et al. 2017 shows measurements of subglacial melt rate using Worldview images. Using mass conservation in a Lagrangian framework, they measure the spatial pattern of melt rate and conclude that the ice shelf Nioghalvfjerdsfjorden is losing mass while the two others, Petermann and Ryder, are still close from equilibrium.

The paper is well written. The use of high resolution stereo images such as Worldview to derive melt rate is good, and the results on the melt rate and mass balance of the ice shelves are interesting and important for the community. This paper should be published after revisions of the comments below.

First of all, I think that the title could be slightly different. I do not like the term "remain-

[Figure]

ing" without a date. The title also suggests that you are looking at all the ice shelves in Greenland, but the manuscript does not study Steensby, Ostenfeld, Hagen Brae, Academy or Zachariae Isstrom (I agree that they are much smaller). In addition, the date of the study (2011-2015) should be included in the title.

Secondly, more details on when exactly the Worldview data were acquired are needed and the authors should explain better how they are merged. The readers know that you use data acquired between 2011 and 2015, but were the images acquired only in years 2011 and 2015, or continuously during the 4/5 years? I believe that Worldview are rather small in coverage. Did the authors rely on different stereo images acquired at different time periods to cover entirely the ice shelves? The authors should provide (could be as supplemental material) a table and maps showing the time tags of the data combined here. They should also state that they assume no change in melt rate between 2011 and 2015.

Another point that would need more details is the distinction between surface and submarine melt rates. It would help the readers if the authors add few lines explaining how they separate the two, most probably using RACMO2.3. Later in the text (line 6, page 6), the authors estimate the annual mean surface melt rates using RACMO2.3, providing one value per ice shelf. Did the authors compute a single mean value for each ice shelves or did they subtract the mean 2011-2015 map of surface melt (which is not spatially uniform) from the total melt? It would also be interesting to know if the authors used an average surface melt over the period 2011-2015, another period, or directly subtract for each pair of worldview images using the exact surface melt between the two acquisition dates. I agree that this correction might be small compared to submarine melt, but it would definitively give stronger results if this is done and explained properly.

line 5: The authors should mention which densities (ice, sea water) were used to translate the surface elevation into ice thickness.

line 9: Recent published grounding line from Mouginot et al. 2015 could also have been used for 79N.

line 12: Would it make sense to use a firn model to refine the ice thickness estimate? I believe that RACMO has published this type of products.

Figure 1: Is it really the submarine melt or total melt including SMB?

Table 1: The authors mention that the uncertainties on the submarine melt rates are derived from surface elevation. It should also include uncertainties on the SMB. As stated previously, the manuscript would really benefit for a better description on the separation between submarine and surface melt rates.

---

## Referee Comment (RC2) · Anonymous Referee #2 · 30 Jul 2017

This manuscript presents an analysis of ice shelf melt rates and its spatial distribution from an interesting database of high resolution surface elevation maps over the three main ice shelves in Greenland. The manuscript is well written, and apart from method-ological details that need to be addressed is of high interest to the community. So I recommend publication after relatively minor revisions.

Methodology and relevant details to be incorporated:

- for each ice shelf: temporal distribution of DEMs (one full DEM per year?); is there really so many DEMs that temporal aliasing issues are irrelevant? Could you please add more details on the subject?

- how are the time average computed? As the mean of monthly binned differences?

[Figure]

- where is the result from one difference between 2 scenes assumed to reside? At one location between the start and end point, following a streamline? At the starting point? Does that choice impact the end result?

- as noted by other users or proponents of this methodology, it may rely (or not) on a few assumptions, including uniform firn density (both spatially and temporally), ice density (idem), and hydrostasy (negligible bridge stresses). You do mention bridge stresses, but seem to gloss over others (e.g. see below). Can you please elaborate on those other assumptions and their eventual impacts on your results?

Other comments:

Page 2, Line 22: remove comma before Falkner ref.

Page 3, Line 24: Reference? I hear this statement a lot, and it does make some sense, but how correct is it?

Page 5, Line 19: I agree, but at the very least you are trying to look at the spatial distribution of melt, so stating that hydrostasy works at the entire ice shelf scale does not really help your case. I understand that quantifying errors resulting from methodological assumptions is difficult, but you probably don't want to underestimate the impacts of such assumptions.

Page 6, Line 5: an independent source of quantification for PG channelized melt can be found in [Dutrieux et al., 2014], with some values that appear to be broadly consistent with yours.

Page 6, Line 21-24: This is all rather speculative. And this raises an important point: that of the spatial distribution of melt and its temporal variation, especially if a major change in the geometry like a calving event occurs. Maybe move to the discussion? Or at least attempt to clarify?

Page 6, Line 25: Ok. But is there a cross-correlation between draft and slope? If so (or not), those are not independent parameters, and may be worth noting.

Page 7, Line 16: What do you mean by 'heterogeneity'? And also, how do your methodological assumptions impact the channel melt signal? Would you expect it to be smoothed? Enhanced? Can you trust it?

Your conclusion section/paragraph:

Can one use a 4-year record and deduce climatic trends? Shouldn't one expect to see temporal variability of melt? And if so do we know how this melt signal translate to ice dynamics? I would agree that there is a potential for a dynamical response if all things were stable in time, but they probably won't, and you may have sampled a particular time period, or not. So I think readers would benefit from a statement on the numerous possibilities ahead here.

Dutrieux, P., C. Stewart, A. Jenkins, K. W. Nicholls, H. F. J. Corr, E. Rignot, and K. Steffen (2014), Basal terraces on melting ice shelves, Geophys. Res. Lett., 41(15), 5506–5513, doi:10.1002/2014GL060618. [online] Available from: http://doi.wiley.com/10.1002/2014GL060618

---

## Author Comment (AC1) · 20 Sep 2017

We thank the two anonymous reviewers for their comments which make a number of important points and provide to several helpful suggestions. We have responded to these below, and we think the proposed changes improve the manuscript under consideration.

**Title choice**

Reviewer 1 points out that using the term "remaining" without a specific date and without consideration of a number of smaller but still existing ice tongues is not entirely correct. We propose the more-specific title "*Satellite-derived submarine melt rates and mass balance (2011–2015) for Greenland's largest remaining ice tongues.*"

[Figure]

[Figure]

**WorldView data description**

Both reviewers point out that information regarding the data used is too lacking.

In all, we use a total of 108 DEMs for Nioghalvfjerdsbræ, 97 DEMs for Petermann Glacier, and 36 DEMs for Ryder Glacier. A table has been added to the manuscript including this information (*Methodology*), and a list indicating the scene IDs for all DEMs used can be made available as supplementary information. The acquisition dates of these DEMs is scattered over the period 2011–2015, however there is a seasonal bias as optical imagery is acquired during the melt season. The spatial distribution of the DEMs is not uniform, as shown in the attached maps below (Figure 1).

From these DEMs, we construct 915 DEM pairs for Nioghalvfjerdsbræ, 751 for Petermann Glacier, and 211 for Ryder Glacier. The temporal distribution of these spans is summarized in the Figure 2.

We propose adding these figures and a description to a supplementary materials section.

**Mass balance clarifications**

As brought up by reviewer 1,

"Another point that would need more details is the distinction between surface and submarine melt rates. It would help the readers if the authors add few lines explaining how they separate the two, most probably using RACMO2.3. Later in the text (line 6, page 6), the authors estimate the annual mean surface melt rates using RACMO2.3, providing one value per ice shelf. Did the authors compute a single mean value for each ice shelves or did they subtract the mean 2011–2015 map of surface melt (which is not spatially uniform) from the total melt? It would also be interesting to know if the authors used an average surface melt over the period 2011–2015, another period, or directly subtract for each pair of worldview images using the exact surface melt between the two acquisition dates. I agree that this correction might be small compared to submarine

melt, but it would definitively give stronger results if this is done and explained properly."

We have expanded the manuscript with the following information:

The surface component of the mass balance is extracted from the RACMO2.3 (Noël et al., 2015) model product. We use a single average melt rate computed over the period 2011–2015 for each ice tongue, which is appropriate for mean melt rates computed over multiple years.

**Ice and seawater densities**

Both reviewers request more information about the densities assumed. To the *Methodology* section of the manuscript we have now included and assumed ice column-averaged density of $920\,\mathrm{kg\,m^{-3}}$, and for seawater, $1028\,\mathrm{km\,m^{-3}}$.

This work does not build or incorporate a firn model, and firn properties are assumed stationary. We have made this assumption explicit in *Errors and uncertainties*. Reviewer 2 writes

"as noted by other users or proponents of this methodology, it may rely (or not) on a few assumptions, including uniform firn density (both spatially and temporally), ice density (idem), and hydrostasy (negligible bridge stresses). You do mention bridge stresses, but seem to gloss over others (e.g. see below). Can you please elaborate on those other assumptions and their eventual impacts on your results?"

Regarding firn density, we assume no variation in time or space along the ice tongues. The section *Errors and uncertainties* has been amended with the note

We expect [firn-densification] to be small as the ice tongues considered are well below the equilibrium line and any remaining firn layer is expected to be thin and assumed constant. This is supported by field data described by Dutrieux et al. (2014) indicating firn compaction (at PG) to be negligible over a one year period.

If significant firn compaction does occur along the ice tongue flow line, our melt rate

estimates would tend to over estimate downstream submarine melting, making our conclusions about the concentration of melt at the grounding line stronger.

We make a similar assumption for ice density. Again, densification downstream would result in an increased the melt rate gradient in our estimates. Snow presents a difficulty and prevents us from making conclusions about seasonal melt variations. Averaged over a year or several years density variations from melting or accumulating snow layers are assumed to cancel, which requires there to be no bias introduced by a regular trend in snow depth along the flow line.

**Time averages**

Reviewer 2 asks,

"how are the time average computed? As the mean of monthly binned differences?"

Time averages are computed by aggregating a monthly means, described in *Methodology: Melt rates*. We modify the text to make it clear that we are averaging over monthly means.

We compute a temporal mean by averaging over average monthly binned estimates of $Dh/Dt$ (January–December) in order to offset bias due to the optical imagery being more available during seasons with more daylight.

As the DEM pairs used extend over multiple months, we weight each month according to the fraction of the month contained within each time span and use the weight to distribute the contribution of each month to submarine melting.

**Spatial identification of melt**

Reviewer 2 asks

"where is the result from one difference between 2 scenes assumed to reside? At one location between the start and end point, following a streamline? At the starting point? Does that choice impact the end result?"

When calculating the Lagrangian ice thickness change, we simplify by assigning the computed difference to the starting point of the streamline. In reality, we expect that the melting is distributed over the entire streamline. This information could be used in the averaging and interpolation of melt rates, or a better approximation assigned the melt to some midpoint along the streamline could be made. To estimate the effect that this improvement would have we consider scenarios in the two following contexts:

- a point in the center of Nioghalvfjerdsbræ

- a point near the grounding line of Nioghalvfjerdsbræ

For a central point on Nioghalvfjerdsbræ, a typical surface ice velocity estimated from our data is $650\,\mathrm{m\,yr}^{-1}$. For a one-year temporal baseline, this would shift the estimate in Figure 1 downstream by just over one $512\,\mathrm{m}$ pixel. For a point at the grounding line, where surface velocities are closer to $1.3\,\mathrm{km\,yr}^{-1}$, the correction would be approximately two pixels. We think based on these examples that at these scales the difference in general is not large.

**Miscellaneous comments, reviewer 1**

1. Figure 1 shows our computed submarine melt rates, i.e. the caption is correct.

2. In Table 1, the volume fluxes do include estimates of the surface mass balance uncertainty, which is added to the $Dh/Dt$ uncertainty to arrive at the final submarine melt error. This can be made more explicit in the text and caption. The uncertainty chosen ($0.8\,\mathrm{m.w.e\,yr}^{-1}$) at all locations was estimated from Table 4 in Noël et al. (2015). These measurements (made along the K-transect in western Greenland) are far from the ice tongues considered.

**Miscellaneous comments, reviewer 2**

1. (Re: accidental comma) Corrected.

2. (Re: reference for statement that velocity is depth-independent) If the assumption of low lateral drag is permitted, then Weertman (1957) is an appropriate reference for this statement.

3. (Re: hydrostasy) Yes, although we have tried to make the case that hydrostasy may be reasonable at low resolutions that are nonetheless higher than the full ice tongue scale. Proper quantification of this error is non-trivial and requires an in depth analysis, as the degree to which any portion of the ice tongue is out of hydrostatic equilibrium depends on the spatial scale of thickness variations, the local melt rate history, and over which the ice has relaxed toward hydrostasy. While not the rigorous analysis that this problem deserves, we could point to modelling in Drews (2015) in which they show that under a specific set of circumstances, a ~1 km channel in 280 m thick ice is nearly in balance while a ~1/2 km channel is not.

4. (Re: PG channelization (Dutrieux, 2014)) Thank you for reminding us of this work.

5. (Re: PG pre-calving melt rates) Moving this to the discussion is one possibility, but breaking it up seems to harm clarity. Instead, we have rephrased this to de-emphasize it as a result and to ground it in what are safe assumptions ("...we do not observe submarine melting at PG to be a driver of substantial net volume loss in its current configuration. However, based on a conservative estimate of melting under the former terminus of PG of 5–10 m yr$^{-1}$ and a calved area of approximately 250 km$^2$, we estimate that the pre-2010 melt flux may have been around 13 km$^3$ yr$^{-1}$ w.e. It is therefore possible that the imbalance between melting and advective replenishment was greater prior to 2010.").

6. (Re: non-independence of draft and slope) Certainly, and this should be noted. We include minor revisions of the text to make this clear.

7. (Re: channelization and heterogeneity) By heterogeneity, we attempt to summarize results in Sergienko (2013) indicating that across-glacier variability in flow and upstream geometry can lead to the development of channels. Our comment here may not add meaningfully to the discussion, and has been replaced with a discussion of methodological limitations. In terms of methodological constraints on detecting melting, we are limited by imperfect knowledge of the surface mass balance (drifting snow settling in surface depressions would mute the measured melt rate) and hydrostasy (bridging effects in regions of rapid melting would also mute the measured melt rate).

8. (Re: conclusion) We absolutely agree on the limitations of a four year record in drawing conclusions about climate, and would like to re-emphasize that these final clauses are speculative based on available data. The dynamical response of the grounding ice sheet to a melt signal is not a satisfactorily resolved question either, resulting in large uncertainties. We have slightly adjusted the *Conclusion* to make it clear that we recognize the limitations in our data for making inferences about climate ("While our mass deficit estimates are based on an average over a relatively short (4 year) time span, high rates of mass loss lead us to speculate that major changes will take place at 79N in the future as the ice tongue thins and eventually becomes ungrounded at its terminus.").

**References**

B Noël, WJ Van De Berg, E Van Meijgaard, P Kuipers Munneke, R Van De Wal, and MR Van Den Broeke. Evaluation of the updated regional climate model RACMO2.3: summer snowfall impact on the Greenland Ice Sheet. *The Cryosphere*, 9(5):1831–1844, 2015. doi: 10.5194/tc-9-1831-2015.

Pierre Dutrieux, Jan De Rydt, Adrian Jenkins, Paul R. Holland, Ho Kyung Ha, Sang Hoon Lee, Eric J. Steig, Qinghua Ding, E. Povl Abrahamsen, and Michael Schröder. Strong

sensitivity of Pine Island ice-shelf melting to climatic variability. *Science*, 343(6167): 174–178, 2014. ISSN 0036-8075. doi: 10.1126/science.1244341.

J. Weertman. Deformation of floating ice shelves. *Journal of Glaciology*, 3(21):38–42, 1957. doi: 10.1017/S0022143000024710.

R. Drews. Evolution of ice-shelf channels in Antarctic ice shelves. *The Cryosphere Discussions*, 9(2):1603–1631, 2015. doi: 10.5194/tcd-9-1603-2015.

O. V. Sergienko. Basal channels on ice shelves. *Journal of Geophysical Research: Earth Surface*, 118(3):1342–1355, August 2013. ISSN 2169-9003. doi: 10.1002/jgrf.20105.

Please also note the supplement to this comment:
https://www.the-cryosphere-discuss.net/tc-2017-99/tc-2017-99-AC1-supplement.pdf

[Figure]

**Fig. 1.**

[Figure]

**Fig. 2.**

**Supplement:**

Table 1: List of all WorldView scene pairs used to construct DEMs

| Scene 1 | Scene 2 | Acquisition date | Ice tongue |
|---|---|---|---|
| 103001000A38B000 | 103001000A50E700 | 2011-04-05 | 79N |
| 103001000A219600 | 103001000A728C00 | 2011-04-05 | 79N |
| 103001000A229F00 | 1030010009153100 | 2011-04-07 | PG |
| 103001000AD56900 | 103001000A1B0E00 | 2011-04-07 | PG |
| 103001000A878400 | 10300100099B7A00 | 2011-04-11 | 79N |
| 103001000AC52800 | 103001000A8BD400 | 2011-04-11 | 79N |
| 1020010012AEC700 | 102001001070E000 | 2011-04-11 | 79N |
| 103001000A95A600 | 103001000A626900 | 2011-04-11 | PG |
| 103001000A010200 | 103001000A49BF00 | 2011-04-11 | PG |
| 1030010009B82900 | 103001000A704F00 | 2011-04-12 | PG |
| 103001000A6ACB00 | 103001000A961E00 | 2011-04-12 | PG |
| 103001000AC59500 | 103001000A904D00 | 2011-04-12 | PG |
| 10200100116D3400 | 102001001201DD00 | 2011-04-26 | 79N |
| 1020010013B66C00 | 102001001223E500 | 2011-04-26 | 79N |
| 102001001337A800 | 1020010013E1B900 | 2011-05-31 | 79N |
| 1020010013738000 | 1020010013A86F00 | 2011-06-01 | 79N |
| 103001000B826F00 | 103001000B2A4400 | 2011-06-01 | PG |
| 102001001458A400 | 1020010014460500 | 2011-06-08 | 79N |
| 1020010012842600 | 102001001331F400 | 2011-06-08 | 79N |
| 103001000B05D700 | 103001000B5C1D00 | 2011-06-09 | 79N |
| 103001000BD86D00 | 103001000B9F0B00 | 2011-06-11 | PG |
| 103001000BBD3F00 | 103001000B71F500 | 2011-06-11 | PG |
| 103001000B065200 | 103001000B9FF600 | 2011-06-12 | PG |
| 103001000B966D00 | 103001000A493A00 | 2011-06-12 | PG |
| 103001000BC5F600 | 103001000B983F00 | 2011-06-12 | PG |
| 1020010013B98000 | 1020010012DFD600 | 2011-06-13 | PG |
| 1020010014644A00 | 1020010014B06A00 | 2011-06-18 | PG |
| 10200100145E7300 | 1020010015B4C600 | 2011-07-11 | PG |
| 103001000CCDB200 | 103001000BCC5000 | 2011-07-12 | PG |
| 103001000C1FFB00 | 103001000CC61200 | 2011-07-12 | PG |
| 1020010013759900 | 10200100140A9400 | 2011-07-15 | PG |
| 1020010015509A00 | 1020010014C80C00 | 2011-07-16 | 79N |
| 1020010014C66E00 | 1020010013D73300 | 2011-07-17 | 79N |
| 102001001548DA00 | 1020010015CB3500 | 2011-08-25 | 79N |
| 10200100151B8600 | 10200100155F2700 | 2011-08-25 | 79N |
| 1020010019D4EC00 | 102001001908B300 | 2012-03-17 | 79N |
| 1030010012838700 | 1030010012AD8300 | 2012-03-26 | PG |
| 103001001234D500 | 1030010012A50A00 | 2012-04-17 | PG |
| 10300100129CAA00 | 1030010013D28E00 | 2012-04-17 | 79N |
| 1030010012816700 | 1030010013109E00 | 2012-04-17 | 79N |
| 102001001AC08300 | 102001001AE14F00 | 2012-04-19 | PG |
| 1030010017442400 | 1030010017B34300 | 2012-04-27 | 79N |

| Scene 1 | Scene 2 | Acquisition date | Ice tongue |
|---|---|---|---|
| 10200100198A9500 | 1020010018D64700 | 2012-04-30 | 79N |
| 10300100197E1400 | 1030010018054500 | 2012-05-10 | RG |
| 10300100187D6D00 | 10300100186FC200 | 2012-05-10 | RG |
| 1030010019089A00 | 10300100188E8E00 | 2012-05-10 | RG |
| 102001001B90E400 | 102001001BC6F900 | 2012-05-10 | RG |
| 1030010019A8FA00 | 103001001964FE00 | 2012-05-12 | 79N |
| 102001001B16F200 | 102001001A977A00 | 2012-05-13 | 79N |
| 102001001A7F8100 | 102001001BE40F00 | 2012-05-14 | 79N |
| 1030010019CEF300 | 103001001923F100 | 2012-05-15 | 79N |
| 10300100187CEE00 | 103001001912FE00 | 2012-05-15 | 79N |
| 1030010018BB3D00 | 1030010019C89F00 | 2012-05-15 | PG |
| 103001001847B400 | 1030010019A28B00 | 2012-05-17 | 79N |
| 103001001826E500 | 1030010018413400 | 2012-05-18 | PG |
| 102001001AD74800 | 102001001A1C9300 | 2012-05-23 | 79N |
| 102001001B8BD000 | 102001001B48E800 | 2012-05-23 | 79N |
| 102001001AC86C00 | 102001001B7FF500 | 2012-05-24 | RG |
| 103001001958C200 | 1030010018C96700 | 2012-05-24 | 79N |
| 103001001817A400 | 103001001894C500 | 2012-05-24 | 79N |
| 102001001B772400 | 102001001B530900 | 2012-05-31 | 79N |
| 102001001B487300 | 102001001A194A00 | 2012-06-02 | PG |
| 102001001B348A00 | 102001001AA65A00 | 2012-06-21 | 79N |
| 103001001BC5CD00 | 103001001A9FAA00 | 2012-07-30 | 79N |
| 102001001D5C0C00 | 102001001C110B00 | 2012-08-22 | PG |
| 1030010020779A00 | 103001002065CB00 | 2013-03-11 | PG |
| 1020010022927600 | 1020010022C1C100 | 2013-03-22 | RG |
| 102001002017E400 | 1020010020073300 | 2013-03-28 | PG |
| 102001001F08DE00 | 1020010020612900 | 2013-03-28 | PG |
| 1020010021CA1000 | 1020010021131A00 | 2013-03-28 | PG |
| 10300100206DBE00 | 10300100206DCE00 | 2013-03-28 | PG |
| 103001002103E200 | 1030010021279200 | 2013-04-01 | PG |
| 1020010022C87200 | 1020010022D4A500 | 2013-04-02 | 79N |
| 10300100218AAC00 | 103001002166FC00 | 2013-04-03 | PG |
| 10300100215DCC00 | 103001002095E700 | 2013-04-03 | PG |
| 10300100212D8600 | 10300100229C6A00 | 2013-04-03 | PG |
| 1030010020347C00 | 10300100201D8900 | 2013-04-04 | RG |
| 102001002055DF00 | 102001002156EC00 | 2013-04-06 | PG |
| 1030010021894600 | 10300100214A4E00 | 2013-04-15 | RG |
| 1030010021982B00 | 10300100208AEB00 | 2013-04-19 | 79N |
| 1030010022366B00 | 1030010022326900 | 2013-04-20 | 79N |
| 1020010022A92B00 | 1020010020333200 | 2013-04-20 | 79N |
| 1030010021B10A00 | 1030010021584300 | 2013-04-23 | 79N |
| 102001002148C500 | 1020010022D17600 | 2013-04-24 | 79N |
| 10300100224F0E00 | 1030010022105300 | 2013-04-26 | PG |
| 1030010022A51F00 | 1030010020420600 | 2013-05-01 | 79N |
| 1030010022905800 | 1030010022279D300 | 2013-05-05 | 79N |

| Scene 1 | Scene 2 | Acquisition date | Ice tongue |
|---|---|---|---|
| 1030010022D00B00 | 1030010023198700 | 2013-05-05 | 79N |
| 10300100228B0100 | 103001002088C00 | 2013-05-10 | PG |
| 1020010021603E00 | 1020010021BF3800 | 2013-05-10 | 79N |
| 1030010022541D00 | 10300100223CF900 | 2013-05-20 | PG |
| 1020010024638D00 | 1020010022299900 | 2013-05-22 | 79N |
| 10300100226FE400 | 1030010023B24100 | 2013-05-24 | 79N |
| 1020010022489600 | 1020010020871C00 | 2013-05-24 | 79N |
| 10200100237A7300 | 10200100212B6C00 | 2013-05-25 | 79N |
| 1020010024CEC700 | 1020010021D7BE00 | 2013-05-26 | 79N |
| 103001002426D600 | 1030010022ADD800 | 2013-05-26 | 79N |
| 1020010022069600 | 1020010021CD2300 | 2013-05-26 | 79N |
| 1030010022A95300 | 1030010023BC9900 | 2013-05-27 | 79N |
| 1030010022D61300 | 10300100235CD200 | 2013-05-27 | 79N |
| 10300100222A5300 | 10300100219D7900 | 2013-05-28 | 79N |
| 103001002365A500 | 1030010023887700 | 2013-05-28 | 79N |
| 102001002174F400 | 1020010022B2E100 | 2013-05-29 | 79N |
| 1030010022368800 | 1030010023238900 | 2013-05-29 | 79N |
| 10200100240DD300 | 10200100230AE100 | 2013-05-31 | PG |
| 103001002390DF00 | 10300100234AF200 | 2013-06-27 | PG |
| 10300100244BA500 | 1030010024AC7900 | 2013-06-28 | RG |
| 1020010022307400 | 102001002321AC00 | 2013-07-11 | PG |
| 1030010025150700 | 1030010025D66E00 | 2013-07-11 | RG |
| 10200100231CC500 | 1020010023E23400 | 2013-07-19 | 79N |
| 1030010024723400 | 103001002483B000 | 2013-07-26 | 79N |
| 1020010024AA0300 | 102001002386E900 | 2013-07-29 | 79N |
| 10300100248ECF00 | 1030010025734B00 | 2013-07-29 | 79N |
| 1030010025119200 | 1030010025B3E300 | 2013-08-01 | 79N |
| 1030010025360100 | 1030010024801900 | 2013-08-04 | 79N |
| 10300100241C1D00 | 1030010026179800 | 2013-08-04 | PG |
| 102001002243FF00 | 1020010024D89A00 | 2013-08-06 | 79N |
| 1020010024B7B400 | 1020010025461C00 | 2013-08-10 | 79N |
| 1020010023B84000 | 1020010023769E00 | 2013-08-10 | 79N |
| 1030010025528300 | 1030010025D3D500 | 2013-08-20 | 79N |
| 103001002665B800 | 10300100268A0400 | 2013-08-21 | PG |
| 10300100269FCD00 | 1030010026949C00 | 2013-08-21 | PG |
| 1020010024C54200 | 1020010024E79B00 | 2013-08-21 | 79N |
| 1020010023DB5E00 | 1020010021062E00 | 2013-08-23 | 79N |
| 1030010025892400 | 10300100275D2100 | 2013-09-06 | 79N |
| 10300100261E0000 | 1030010026306600 | 2013-09-08 | PG |
| 103001002790A300 | 10300100287D8200 | 2013-09-13 | PG |
| 1020010024AA5600 | 1020010025577A00 | 2013-09-23 | RG |
| 1020010024E0BF00 | 10200100250FA600 | 2013-10-03 | PG |
| 10300100279E6200 | 1030010028AE6C00 | 2013-10-05 | PG |
| 1030010027D25D00 | 103001002726B100 | 2013-10-08 | 79N |
| 103001002D7E7600 | 103001002EC1C200 | 2014-03-27 | PG |

| Scene 1 | Scene 2 | Acquisition date | Ice tongue |
|---|---|---|---|
| 103001002E791500 | 10300100303A3100 | 2014-04-04 | 79N |
| 102001002DC0DF00 | 102001002CAEC500 | 2014-04-05 | 79N |
| 102001002D057F00 | 102001002E675800 | 2014-04-06 | 79N |
| 1030010030AA1D00 | 103001002F63F100 | 2014-04-08 | 79N |
| 103001002F565400 | 103001002F171B00 | 2014-04-12 | 79N |
| 103001002E917100 | 103001002E113600 | 2014-04-13 | PG |
| 103001002E5A0300 | 103001002F0F7F00 | 2014-04-16 | 79N |
| 103001002F0BED00 | 103001003088B400 | 2014-04-18 | PG |
| 1030010030616000 | 103001002E527800 | 2014-04-22 | 79N |
| 102001002CD2F700 | 102001002D4BA300 | 2014-04-24 | 79N |
| 103001002FC3DF00 | 103001002F9A3300 | 2014-04-25 | PG |
| 1030010030723600 | 103001003053D200 | 2014-04-28 | PG |
| 102001002D347E00 | 102001002BA28700 | 2014-04-28 | PG |
| 1030010030535B00 | 103001003149500 | 2014-04-30 | PG |
| 1030010030976700 | 1030010031C61200 | 2014-04-30 | 79N |
| 103001002F731300 | 10300100305FF300 | 2014-05-01 | PG |
| 102001002E5F6100 | 102001002EA42000 | 2014-05-01 | 79N |
| 1030010030115F00 | 1030010031563600 | 2014-05-03 | 79N |
| 10300100308AF800 | 103001003057C400 | 2014-05-04 | 79N |
| 102001002C300300 | 102001002C1B5D00 | 2014-05-07 | 79N |
| 1030010031B51300 | 1030010030B49E00 | 2014-05-08 | 79N |
| 103001002FB40800 | 103001002F36C800 | 2014-05-08 | 79N |
| 1030010031882100 | 1030010031AAD400 | 2014-05-12 | PG |
| 102001002E070100 | 102001002F5D4200 | 2014-05-17 | 79N |
| 1030010030402000 | 1030010031687B00 | 2014-05-19 | PG |
| 102001002F22C000 | 102001002E5C9100 | 2014-05-19 | PG |
| 1030010032287200 | 1030010032788A00 | 2014-05-23 | PG |
| 1030010031C96D00 | 1030010032934A00 | 2014-05-26 | RG |
| 1030010031A95C00 | 1030010032061A00 | 2014-05-28 | PG |
| 1030010031BC2400 | 10300100300C8C00 | 2014-05-29 | RG |
| 1030010032D67600 | 1030010031D2A100 | 2014-05-29 | 79N |
| 102001002F5AFC00 | 102001002E77E000 | 2014-05-31 | RG |
| 102001002FD5BD00 | 102001002E041C00 | 2014-05-31 | PG |
| 102001002EE1B600 | 102001003022BD00 | 2014-06-01 | 79N |
| 102001002FEA4E00 | 102001002FE18200 | 2014-06-01 | RG |
| 102001002F5FF800 | 102001002E833400 | 2014-06-01 | PG |
| 102001002F523600 | 102001002E26F600 | 2014-06-04 | PG |
| 10300100333B9F00 | 103001032415D00 | 2014-06-17 | 79N |
| 1030010033A2D400 | 1030010335D2400 | 2014-06-20 | PG |
| 1020010030907500 | 1020010031C91500 | 2014-06-20 | PG |
| 1030010032543300 | 1030010032026400 | 2014-06-22 | PG |
| 102001002F301800 | 102001002FC47C00 | 2014-06-23 | PG |
| 103001003308BC00 | 103001003256F800 | 2014-06-24 | PG |
| 102001002F860600 | 102001003086C700 | 2014-06-27 | PG |
| 102001002E349300 | 10200100318E0200 | 2014-06-29 | PG |

| Scene 1 | Scene 2 | Acquisition date | Ice tongue |
|---|---|---|---|
| 10300100340AF800 | 10300100332FD900 | 2014-06-30 | PG |
| 1020010031E8AD00 | 102001002EAAA300 | 2014-07-04 | PG |
| 1030010033713E00 | 103001003376F100 | 2014-07-06 | PG |
| 1030010033969600 | 10300100336CFF00 | 2014-07-06 | RG |
| 1030010033916300 | 10300100330C6600 | 2014-07-07 | 79N |
| 103001003405BE00 | 10300100337C6C00 | 2014-07-07 | RG |
| 103001003408280 | 103001003265C300 | 2014-07-16 | PG |
| 1030010035D14700 | 103001034735900 | 2014-08-04 | RG |
| 102001003253690 | 102001031302200 | 2014-08-06 | RG |
| 102001003231EA00 | 102001003272FB00 | 2014-08-09 | PG |
| 102001003521BA00 | 1020010032934D00 | 2014-08-11 | RG |
| 10300100357B5000 | 10300100369C2A00 | 2014-08-12 | PG |
| 1020010031030D00 | 1020010032B0C600 | 2014-08-15 | RG |
| 1020010030ABC300 | 10200100329BDB00 | 2014-08-15 | PG |
| 10300100357F6600 | 10300100347ABF00 | 2014-08-16 | PG |
| 1020010030B9A400 | 102001003399BB00 | 2014-08-17 | RG |
| 10300100352C8300 | 10300100363CD400 | 2014-08-18 | PG |
| 1020010033A3B000 | 1020010032EA1100 | 2014-08-18 | RG |
| 1020010031C81400 | 1020010031B27F00 | 2014-08-22 | RG |
| 102001003464C00 | 1020010033D30300 | 2014-08-24 | PG |
| 10300100357D8800 | 103001034032C00 | 2014-08-25 | RG |
| 103001003605DD00 | 10300100364E9600 | 2014-08-26 | RG |
| 102001003022680 | 1020010031E37300 | 2014-08-26 | PG |
| 1030010035C90200 | 103001035007100 | 2014-08-27 | RG |
| 103001003552310 | 1030010036990C00 | 2014-08-28 | RG |
| 10200100325FCB00 | 1020010031C32C00 | 2014-08-30 | PG |
| 1020010034C73200 | 10200100333DEC00 | 2014-08-31 | PG |
| 1030010036772600 | 10300100374F1700 | 2014-08-31 | RG |
| 103001003522EB00 | 103001003682A000 | 2014-09-01 | PG |
| 1030010036B2F500 | 103001037803800 | 2014-09-04 | RG |
| 1020010032DB6B00 | 102001032585000 | 2014-09-05 | RG |
| 1030010035BD9500 | 1030010036B7BD00 | 2014-09-06 | PG |
| 1020010033C99300 | 1020010034CB4300 | 2014-09-18 | PG |
| 1030010036B6FA00 | 103001036442400 | 2014-09-20 | 79N |
| 103001004177AE00 | 103001003EAD1D00 | 2015-03-14 | 79N |
| 104001000971EA00 | 1040010009BC2000 | 2015-03-15 | 79N |
| 102001003DD47D00 | 102001003B1A2900 | 2015-04-06 | 79N |
| 10400100092E6500 | 104001000A6E9400 | 2015-04-06 | 79N |
| 102001003D519A00 | 102001003D72DE00 | 2015-04-09 | PG |
| 102001003D249200 | 102001003E8DB800 | 2015-04-10 | PG |
| 1020010039C56400 | 102001003C918200 | 2015-04-10 | PG |
| 103001003F8D5500 | 10300100405BF200 | 2015-04-11 | PG |
| 10200100400FAB00 | 102001003E92AB00 | 2015-04-12 | PG |
| 102001003BB33600 | 102001003C8F1200 | 2015-04-15 | PG |
| 102001003A8CC700 | 102001003B393B00 | 2015-04-15 | 79N |

| Scene 1 | Scene 2 | Acquisition date | Ice tongue |
|---|---|---|---|
| 1040010009B1ED00 | 104001000A40EB00 | 2015-04-16 | 79N |
| 103001004149A900 | 1030010041D88800 | 2015-04-16 | 79N |
| 10300100406CA100 | 10300100407CC100 | 2015-04-16 | 79N |
| 1030010040B90500 | 1030010042ABDE00 | 2015-04-16 | 79N |
| 1030010042D7C300 | 103001003F9A2700 | 2015-04-17 | 79N |
| 102001003C929A00 | 102001003B825400 | 2015-04-17 | 79N |
| 103001004045D200 | 10300100405D4C00 | 2015-04-19 | PG |
| 104001000A054400 | 104001000A02D600 | 2015-04-26 | RG |
| 102001003C17F000 | 102001003D0AD300 | 2015-04-28 | RG |
| 10300100418F7F00 | 1030010041BFA900 | 2015-04-28 | RG |
| 104001000B4DB800 | 104001000AA07700 | 2015-05-03 | RG |
| 102001003B7A5800 | 102001003C961D00 | 2015-05-05 | RG |
| 102001003F046B00 | 102001003E171500 | 2015-05-16 | 79N |
| 1030010043C01400 | 103001004089E100 | 2015-05-23 | 79N |
| 104001000DB3A100 | 104001000D5D0C00 | 2015-06-17 | 79N |
| 103001004415A700 | 10300100447C4900 | 2015-07-09 | PG |
| 103001004423F800 | 10300100460A0300 | 2015-07-10 | PG |
| 104001000E823E00 | 104001000E783A00 | 2015-07-17 | 79N |
| 103001004647F000 | 103001004783ID00 | 2015-07-24 | 79N |

---

## Author Response (AR1)

The following includes a point-by-point response to the concerns raised by the two reviewers and a description of changes made to the manuscript. Much of this information is repeated from the author's comments, as I was unsure at that time what should be included in which response.

**Title choice**

*Referee comment*

> First of all, I think that the title could be slightly different. I do not like the term "remaining" without a date. The title also suggests that you are looking at all the ice shelves in Greenland, but the manuscript does not study Steensby, Ostenfeld, Hagen Brae, Academy or Zachariae Isstrom (I agree that they are much smaller). In addition, the date of the study (2011-2015) should be included in the title.

*Author response*

We think these are well-made points.

*Manuscript changes*

We have changed the title to "Satellite-derived submarine melt rates and mass balance (2011–2015) for Greenland's largest remaining ice tongues," qualifying "remaining" with "largest" and adding a date range.

**WorldView data description**

*Referee comment*

> Secondly, more details on when exactly the Worldview data were acquired are needed and the authors should explain better how they are merged. The readers know that you use data acquired between 2011 and 2015, but were the images acquired only in years 2011 and 2015, or continuously during the 4/5 years? I believe that Worldview are rather small in coverage. Did the authors rely on different stereo images acquired at different time periods to cover entirely the ice shelves? The authors should provide (could be as supplemental material) a table and maps showing the time tags of the data combined here. They should also state that they assume no change in melt rate between 2011 and 2015.

*Author response*

As noted in the interactive discussion, we agree with these points. We do mention that the computed values are temporal means, so we don't so much assume no change but instead explicitly report the mean value.

*Manuscript changes*

We have clarified the method we used to average mass balance estimates (Section 2.3). In order to provide more detail available regarding the images used and the quantity of DEMs generated, we have added a table to the Methodology section with DEM counts, and supplementary material with the exact imagery used. Visualizations portraying the temporal and spatial distributions of our data have also been included as supplementary material.

**Mass balance clarifications**

*Referee comment*

> Another point that would need more details is the distinction between surface and submarine melt rates. It would help the readers if the authors add few lines explaining how

they separate the two, most probably using RACMO2.3. Later in the text (line 6, page 6), the authors estimate the annual mean surface melt rates using RACMO2.3, providing one value per ice shelf. Did the authors compute a single mean value for each ice shelves or did they subtract the mean 2011–2015 map of surface melt (which is not spatially uniform) from the total melt? It would also be interesting to know if the authors used an average surface melt over the period 2011–2015, another period, or directly subtract for each pair of worldview images using the exact surface melt between the two acquisition dates. I agree that this correction might be small compared to submarine melt, but it would definitively give stronger results if this is done and explained properly.

*Author response*

At the time that this analysis was performed, we had access to the 11 km mass balance product, which does not lend itself to spatially-variable estimates at each ice tongue.

*Manuscript changes*

We have expanded the manuscript with the following information in Section 2.3:

The surface component of the mass balance is extracted from the RACMO2.3 [1] model product. We use a single average melt rate computed over the period 2011–2015 for each ice tongue, which is appropriate for mean melt rates computed over multiple years.

**Ice and seawater densities**

*Referee comment*

Both reviewers request more information about the densities assumed. Additionally, reviewer 2 writes

as noted by other users or proponents of this methodology, it may rely (or not) on a few assumptions, including uniform firn density (both spatially and temporally), ice density (idem), and hydrostasy (negligible bridge stresses). You do mention bridge stresses, but seem to gloss over others (e.g. see below). Can you please elaborate on those other assumptions and their eventual impacts on your results?

*Author response*

This work does not build or incorporate a firn model, and firn properties are assumed stationary in time and space. If significant firn compaction does occur along the ice tongue flow line, our melt rate estimates would tend to over estimate downstream submarine melting, making our conclusions about the concentration of melt at the grounding line stronger.

We make a similar assumption for ice density. Again, densification downstream would result in an increased the melt rate gradient in our estimates. Snow presents a difficulty and prevents us from making conclusions about seasonal melt variations. Averaged over a year or several years density variations from melting or accumulating snow layers are assumed to cancel, which requires there to be no bias introduced by a regular trend in snow depth along the flow line.

*Manuscript changes*

We have made the constant firn assumption explicit in section 2.5.

To the section 2.3 of the manuscript we have now included and assumed ice column-averaged density of $920 \, \mathrm{kg \, m^{-3}}$, and for seawater, $1028 \, \mathrm{km \, m^{-3}}$.

The section 2.5 (Errors and Uncertainties) has been amended with the note

We expect [firn-densification] to be small as the ice tongues considered are well below the equilibrium line and any remaining firn layer is expected to be thin and assumed constant. This is supported by field data described by Dutrieux et al. [2] indicating firn compaction (at PG) to be negligible over a one year period.

**Time averages**

*Referee comment*

Reviewer 2 asks,

how are the time average computed? As the mean of monthly binned differences?

*Author response*

Time averages are computed by aggregating a monthly means, described in section 2.3.

*Manuscript changes*

We modify the text to make it clear that we are averaging over monthly means.

We compute a temporal mean by averaging over average monthly binned estimates of $Dh/Dt$ (January–December) in order to offset bias due to the optical imagery being more available during seasons with more daylight.

As the DEM pairs used extend over multiple months, we weight each month according to the fraction of the month contained within each time span and use the weight to distribute the contribution of each month to submarine melting.

**Spatial identification of melt**

*Referee comment*

Reviewer 2 asks

where is the result from one difference between 2 scenes assumed to reside? At one location between the start and end point, following a streamline? At the starting point? Does that choice impact the end result?

*Author response*

When calculating the Lagrangian ice thickness change, we simplify by assigning the computed difference to the starting point of the trajectory. In reality, we expect that the melting is distributed over the entire streamline. This information could be used in the averaging and interpolation of melt rates, or a better approximation assigned the melt to some midpoint along the streamline could be made. To estimate the effect that this improvement would have we consider scenarios in the two following contexts:

- a point in the center of Nioghalvfjerdsbræ
- a point near the grounding line of Nioghalvfjerdsbræ

For a central point on Nioghalvfjerdsbræ, a typical surface ice velocity estimated from our data is $650 \, \mathrm{m \, yr^{-1}}$. For a one-year temporal baseline, making the alternative extreme assumption, that melt occurs at the downstream location, our estimate in Figure 1 would shift downstream by just over one $512 \, \mathrm{m}$ pixel. For a point at the grounding line, where surface velocities are closer to $1.3 \, \mathrm{km \, yr^{-1}}$, the correction would be approximately two pixels. We think based on these examples that at these scales

the difference in general is not large.

*Manuscript changes*

We have added a paragraph to section 2.5 describing this source of potentially biased error.

> A source of biased error derives from the assignment of Lagrangian melt rates incurred over a trajectory to a single point. In Figure 1, we assign the melt to the original (upstream) measurement location, which tends to shift the melt estimates slightly upstream in map representations. Based on the ice tongue velocities and temporal baselines used, this upstream bias may be up to approximately 1400 m in the extreme case that all melt actually occurs at the final (downstream) location.

**Miscellaneous comments, reviewer 1**

*Referee comment*

> Is it really the submarine melt or total melt including SMB?

*Author response*

Figure 1 shows our computed submarine melt rates (total melt minus surface melt), i.e. the caption is correct.

*Manuscript changes*

None.

*Referee comment*

> The authors mention that the uncertainties on the submarine melt rates are derived from surface elevation. It should also include uncertainties on the SMB. As stated previously, the manuscript would really benefit for a better description on the separation between submarine and surface melt rates.

*Author response*

I regret to write that in the submitted manuscript and previous author comments, what was treated as the submarine melt uncertainty was in fact the total melt uncertainty. Because of error propagation when combined with surface melt uncertainties, the submarine melt uncertainty is now larger than reported. This has been corrected throughout the manuscript. As a side-effect, because our total melt uncertainty is now computed directly rather than derived from submarine and surface melt rates, our conclusions comparing combined melt rates to grounding line fluxes are slightly strengthened.

In Table 1, the volume fluxes do include estimates of the surface mass balance uncertainty, which is added to the $Dh/Dt$ uncertainty to arrive at the final submarine melt error. This can be made more explicit in the text and caption. The uncertainty chosen ($0.8\,\mathrm{m.w.e\,yr^{-1}}$) at all locations was estimated from Table 4 in [1]. These measurements (made along the K-transect in western Greenland) are far from the ice tongues considered.

*Manuscript changes*

We have updated the uncertainty estimates and revised the caption on (now) Table 2 to make clear what we report.

**Miscellaneous comments, reviewer 2**

*Referee comment*

(Re: reference for claim the velocity is depth-independent)

*Manuscript changes*

We have added a reference to Weertman [3].

*Referee comment*

> at the very least you are trying to look at the spatial dis- tribution of melt, so stating that hydrostasy works at the entire ice shelf scale does not really help your case. I understand that quantifying errors resulting from methodolog- ical assumptions is difficult, but you probably don't want to underestimate the impacts of such assumptions.

*Author response*

Proper quantification of this error is non-trivial and requires an in depth analysis, as the degree to which any portion of the ice tongue is out of hydrostatic equilibrium depends on the spatial scale of thickness variations, the local melt rate history, and period over which the ice has relaxed toward hydrostasy. While not the rigorous analysis that this problem deserves, we could point to modelling in Drews [4] in which they show that under a specific set of circumstances, a $\sim 1$ km channel in $280$ m thick ice is nearly in balance while a $\sim 1/2$ km channel is not.

*Manuscript changes*

To avoid discounting the possibility for additional error, we add "limitations due to the our assumption of hydrostasy should be kept in mind when considering small-scale features in the results."

*Referee comment*

(Re: PG pre-calving melt rates)

> This is all rather speculative. And this raises an important point: that of the spatial distribution of melt and its temporal variation, especially if a major change in the geometry like a calving event occurs. Maybe move to the discussion? Or at least attempt to clarify?

*Author response*

Moving this to the discussion is one possibility, but breaking it up seems to harm readability. Instead, we have rephrased this to de-emphasize it as a result and to ground it in what are safe assumptions.

*Manuscript changes*

> ...we do not observe submarine melting at PG to be a driver of substantial net volume loss in its current configuration. However, based on a conservative estimate of melting under the former terminus of PG of $5$–$10$ m yr$^{-1}$ and a calved area of approximately $250$ km$^2$, we estimate that the pre-2010 melt flux may have been around $13$ km$^3$ yr$^{-1}$ w.e. It is therefore possible that the imbalance between melting and advective replenishment was greater prior to 2010.

*Referee comment*

(Re: non-independence of draft and slope)

> But is there a cross-correlation between draft and slope? If so (or not), those are not independent parameters, and may be worth noting.

*Author response*

We agree that this should be noted.

*Manuscript changes*

We add "Given that draft and slope co-vary" to our discussion.

*Referee comment*

(Re: channelization and heterogeneity)

> What do you mean by 'heterogeneity'? And also, how do your methodological assumptions impact the channel melt signal? Would you expect it to be smoothed? Enhanced? Can you trust it?

*Author response*

By heterogeneity, we attempt to summarize results in Sergienko [5] indicating that across-glacier variability in flow and upstream geometry can lead to the development of channels.

*Manuscript changes*

Our comment here may not add meaningfully to the discussion, and has been replaced with a discussion of methodological limitations.

> Detection of channelization using our methodology is limited by non-hydrostatic behaviour around narrow channels and the potential for drifting snow to accumulate in surface troughs associated with channels, and so those that we do detect tend to be large.

*Referee comment*

(Re: conclusion)

> Can one use a 4-year record and deduce climatic trends? Shouldn't one expect to see temporal variability of melt? And if so do we know how this melt signal translate to ice dynamics? I would agree that there is a potential for a dynamical response if all things were stable in time, but they probably won't, and you may have sampled a particular time period, or not. So I think readers would benefit from a statement on the numerous possibilities ahead here.

*Author response*

We absolutely agree on the limitations of a four year record in drawing conclusions about climate, and would like to re-emphasize that these final clauses are speculative based on available data. The dynamical response of the grounding ice sheet to a melt signal is not a satisfactorily resolved question either, resulting in large uncertainties.

*Manuscript changes*

We have slightly adjusted the conclusion to make it clear that we recognize the limitations in our data for making inferences about climate:

> While our mass deficit estimates are based on an average over a relatively short (4 year) time span, high rates of mass loss lead us to speculate that major changes will take place at 79N in the future as the ice tongue thins and eventually becomes ungrounded at its terminus.

[revised manuscript text omitted]